# First–Principles Investigation of the Structural, Elastic, Electronic, and Optical Properties of α– and β–SrZrS_3_: Implications for Photovoltaic Applications

**DOI:** 10.3390/ma13040978

**Published:** 2020-02-21

**Authors:** Henry Igwebuike Eya, Esidor Ntsoenzok, Nelson Y. Dzade

**Affiliations:** 1Department of Material Science and Engineering, African University of Science and Technology, Km 10 Airport Road, Galadimawa, Abuja F.C.T. 900107, Nigeria; heya@aust.edu.ng (H.I.E.); esidor.ntsoenzok@cnrs-orleans.fr (E.N.); 2CEMHTI–CNRS Site Cyclotron, 3A rue de la Férollerie, 45071 Orléans, France; 3The School of Chemistry, Cardiff University, Cardiff CF10 3AT, Wales, UK

**Keywords:** earth–abundant materials, chalcogenide perovskites, Solar cell, Density Functional Theory, Optoelectronic properties

## Abstract

Transition metal perovskite chalcogenides are attractive solar absorber materials for renewable energy applications. Herein, we present the first–principles screened hybrid density functional theory analyses of the structural, elastic, electronic and optical properties of the two structure modifications of strontium zirconium sulfide (needle–like α–SrZrS_3_ and distorted β–SrZrS_3_ phases). Through the analysis of the predicted electronic structures, we show that both α– and β–SrZrS_3_ materials are direct band gaps absorbers, with calculated band gaps of 1.38, and 1.95 eV, respectively, in close agreement with estimates from diffuse–reflectance measurements. A strong light absorption in the visible region is predicted for the α– and β–SrZrS_3_, as reflected in their high optical absorbance (in the order of 10^5^ cm^−1^), with the β–SrZrS_3_ phase showing stronger absorption than the α–SrZrS_3_ phase. We also report the first theoretical prediction of effective masses of photo-generated charge carriers in α– and β–SrZrS_3_ materials. Predicted small effective masses of holes and electrons at the valence, and conduction bands, respectively, point to high mobility (high conductivity) and low recombination rate of photo-generated charge carriers in α– and β–SrZrS_3_ materials, which are necessary for efficient photovoltaic conversion.

## 1. Introduction

Perovskite materials have attracted significant attention from researchers due to their potential in various applications. In photovoltaic applications, inorganic–organic halide perovskite materials represent a great breakthrough in the development of solar cell materials with the power conversion efficiency rising from 3.6 to about 24%, since the first application in 2009 by Kojima et al. [1]. The recently reported efficiencies are comparable to the most advanced thin film solar cells, such as CdTe, GaAs, as well as the silicon-based solar cells. In addition to this unprecedented improvement in power conversion efficiency are the ease and low-cost of synthesis of the materials, as well as the fabrication of the solar cell device. Notwithstanding the rapid increase in their efficiencies, the toxicity of the lead content and the intrinsic instability of the bulk lead–halide perovskite materials and their interface heterostructures remain significant drawbacks to their large–scale applications and commercialization [2,3,4,5]. These concerns have motivated the search for new absorber materials, that are lead–free but with similar electronic and optical properties as lead–halide perovskite materials. Efforts to replace the Pb in lead halide perovskites with low toxic cations including Sn (II) [6], Ag (I) [7], Bi (III) [8,9], Ti (IV) [10] and Sb (III) [10] have, therefore, received significant attention. These materials yielded non–toxic but low efficiency devices, a maximum of 9% power conversion efficiency (PCE) for the Sn based perovskite. Oxide perovskites on the other hand have proven to have higher stability, due to their resistance to moisture^3^, and have been reported to be promising solar absorbers if engineered to have the band gap matched with the solar spectrum [11,12]. 

Chalcogenide perovskites having the structure ABX_3_ (X = S, Se; A, B = metals with combined valence of 6), have recently been proposed for photovoltaic application [13]. Compared to lead halide perovskites, chalcogenide perovskites materials are more environmentally friendly (component elements are earth–abundant and non–toxic) and possess superior electronic and optical properties, suggesting their potential ideal for low–cost tandem solar cell application. So far, a number of transition metal chalcogenide perovskites have been successfully synthesized experimentally and investigated theoretically [3,13,14,15,16,17]. Sun et al. [3] in their theoretical studies, predicted that the band gaps of CaTiS_3_ (1.0 eV), BaZrS_3_ (1.75 eV), CaZrSe_3_ (1.3 eV), and CaHfSe_3_ (1.2 eV) having the distorted perovskite structure are suitable for making single–junction solar cells. They are generally n–type semiconductors, with thin–films of BaZrS_3_ recently to have carrier densities in the range of 10^19^–10^20^ cm^–3^ [18]. Chalcogenide perovskites also display high optical absorption properties, making them promising materials used in addressing the instability and toxicity issues associated with halide perovskites. A number of chalcogenide perovskites including CaZrS_3_, SrTiS_3_, SrZrS_3_ and BaZrS_3_ have been successfully synthesized by Perera et al. [17] using high temperature sulfurization of their oxide counterparts. A widely tunable bandgap range of 1.73–2.87 eV was reported for these materials. Ju et al. [14] predicted a series of chalcogenide perovskites such as SrSnSe_3_, CaSnS_3_ and SrSnS_3_ with tunable direct bandgaps within the optimal range of 0.9–1.6 eV for single junction solar cell applications. Peng et al. [19] demonstrated, from first–principles density functional theory (DFT) calculations, that the optical transitions near the band edges of chalcogenide perovskites differ from those of their halide counterparts.

Single crystal X–ray diffraction analysis of strontium zirconium sulfide (SrZrS_3_) samples show two structure modifications (needle–like α–SrZrS_3_ and distorted β–SrZrS_3_ phases (Figure 1). The α–and β–SrZrS_3_ phases both crystallize in the orthorhombic crystal structure with space group *Pnma*. The optical band gaps were estimated to be 1.52 and 2.05 eV for the α–SrZrS_3_, and β–SrZrS_3_ phases, respectively, from diffuse–reflectance measurements [16]. Thermal stability investigations by Niu et al. [20] on synthesized α–SrZrS_3_, β–SrZrS_3_, BaZrS_3_, Ba_2_ZrS_4_ and Ba_3_Zr_2_S_7_ show that these chalcogenide perovskites possess excellent thermal stability in air for a temperature range up to 550 °C [16]. The fundamental atomic–level insights into the mechanical and structural characteristics, as well as the electronic and optical features of SrZrS_3_ materials, albeit, poorly understood and inconclusive. Previous theoretical studies by Oumertem et al. have characterized the electronic and thermodynamic properties of the cubic and orthorhombic XZrS_3_ (X = Ba, Sr, Ca) compounds, predicting their band gap in the range of 1.151–1.617 eV [15]. There is, however, no systematic theoretical investigation dedicated to elucidating the structural, elastic, and optoelectronic properties of the α–SrZrS_3_ and β–SrZrS_3_ chalcogenide perovskites, which makes this investigation timely.

In the present study, we do not only report the structural, mechanical, and electronic properties of α–SrZrS_3_ and β–SrZrS_3_ by means of screened hybrid (HSE06) density functional theory methods, but also comprehensively characterized the optical properties and discussed their implications for photovoltaic applications. Based on the calculated optical absorbance, reflectivity, and refractive index, we demonstrate that α– and β–SrZrS_3_ are suitable solar absorber materials for solar cell and other optoelectronic applications.

## 2. Computational Details

The density functional theory (DFT) were performed within periodic boundary conditions as implemented in the Vienna Ab initio Simulation Package (VASP) [21,22]. The Perdew-Burke-Ernzerhof (PBE) generalized gradient approximation (GGA) functional [23] was used for geometry optimizations, while for electronic structures and optical calculations, the screened hybrid functional HSE06 with 25% Hartree−Fock exchange and screening parameter of ω = 0.11 bohr ^−1^ was employed [24]. The projected augmented wave (PAW) method was used to describe the interactions between the valence electrons and the cores [25]. Long–range van der Waals (vdW) were accounted for using of the Grimme (DFT–D3) method [26]. A plane–wave basis, set with a kinetic energy cut–off of 600 eV, was tested to be sufficient to converge the total energy of the α–SrZrS_3_ and β–SrZrS_3_ phases to within 10^−6^ eV and the residual Hellmann–Feynman forces on all relaxed atoms reached 10^−3^ eV Å^−1^. The Brillouin zone of the α–SrZrS_3_ and β–SrZrS_3_ phases was sampled using 5 × 7 × 3, and 5 × 3 × 5 Monkhorst–Pack [27] K–points mesh, respectively. 

The optical properties of α–SrZrS_3_ and β–SrZrS_3_ were determined from the complex dielectric function, ε(ω) = ε_1_(ω) + i ε_2_(ω) within the independent–particle formalism [28], where the the imaginary part of the dielectric function is calculated in the long wavelength q→0 limit as,
(1)$$\begin{array}{cc} \varepsilon_{2}\left(\omega \right)=\varepsilon_{\alpha \beta }^{\left(2\right)}\left(\omega \right) & =\frac{4\pi^{2}e^{2}}{\Omega }\underset{q\to 0}{\lim}\frac{1}{q^{2}}\underset{c,v,k}{\sum }2w_{k}\delta \left(\epsilon_{ck}-\epsilon_{vk}-\omega \right)\\ & \times \langle u_{ck+e_{\alpha }q}|u_{vk}\rangle \langle u_{ck+e_{\beta }q}|u_{vk}\rangle {}^{*} \end{array}$$
where, Ω and $w_{k}\;$are the volume of the primitive cell and k–point weights, respectively. The $\;\epsilon_{ck}$($\epsilon_{vk}$) are **k**–dependent conduction (valence) band energies, $u_{vk},u_{ck}$ are the cell periodic part of the pseudo–wave function and $e_{\alpha ,\beta }$ are the unit vectors along the Cartesian directions. From Kramers–Kronig transformations, the real part of dielectric function can be determined from the relation,
(2)$$\varepsilon_{1}\left(\omega \right)=\varepsilon_{\alpha \beta }^{\left(1\right)}\left(\omega \right)=1+\frac{2}{\pi }P\int_{0}^{\infty }\frac{\varepsilon_{\alpha \beta }^{\left(2\right)}\left(\omega \prime \right)\omega \prime }{\omega^{\prime 2}-\omega^{2}+i\eta }d\omega^{\prime }$$
where P denotes the principle value. A small value of 0.1, which is acceptable for most calculations was used for the complex shift $\left(\eta \right)$ to smoothen the real part of the dielectric function. The optical parameters, such as absorption coefficient, are based on $\varepsilon_{1}$ and $\varepsilon_{2}$. The absorption coefficient $\left(\alpha_{abs}\right)$ is calculated using the following relation:(3)$$\alpha_{abs}=\sqrt{2}\omega {\left(\sqrt{\varepsilon_{1}^{2}\left(\omega \right)+\varepsilon_{2}^{2}\left(\omega \right)}-\varepsilon_{1}\left(\varepsilon \right)\right)}^{1/2}.$$

Local field and excitonic effects have been neglected in the present study as they are not accurately treated in the independent–particle formalism. These effects may be accounted for by using expensive methods, such as Bethe–Salpeter equation (BSE) and time–dependent DFT with proper exchange–correlation kernels [29].

The elastic stiffness constants were calculated using the stress–strain method [30]. The strain (σ) and stress (ε) are related according to Hook’s law by σ_i_ = C_ij_ε_j_, where C_ij_ are the elastic stiffness constants. For an orthorhombic system, there are nine independent elastic constants viz; C11, C22, C33, C12, C13, C23, C44, C55 and C66. These elastic constants are used to predict the bulk modulus (B_V_) and the shear modulus (G_V_). Whereas, Young’s modulus (E) and the Poisson’s ratio (v) are in turn predicted from the calculated bulk and shear moduli. The bulk and shear moduli measure the material’s resistance to uniform compression, and shearing strains, respectively. The bulk and shear moduli were calculated using the Voigt approximation [31] which for orthorhombic structures, can be simplified as follows:(4)$$B_{V}=\frac{1}{9}\left(C_{11}+C_{22}+C_{33}\right)+\frac{2}{9}\left(C_{12}+C_{13}+C_{23}\right)$$
(5)$$G_{V}=\frac{1}{15}\left(C_{11}+C_{22}+C_{33}-C_{12}-C_{13}-C_{23}\right)+\frac{1}{5}\left(C_{44}+C_{55}+C_{66}\right)$$

The Young’s modulus (E) and Poisson’s ratio (v), which are characteristic stiffness properties of a material were obtained using the relations: $E=\frac{9BG}{\left(3B+G\right)}$ and $\nu =\frac{\left(3B-2G\right)}{2(3B+G)}$.

## 3. Results and Discussion

### 3.1. Structural Properties

SrZrS_3_ crystallizes in two structure modifications; α and β–phases, both in the orthorhombic crystal system with space group *Pnma* (Figure 1). The α–SrZrS_3_ has five independent lattice sites; one Sr, one Zr, and three S sites, whereas in the β–phase, there exist only four positions (one Sr, one Zr, and two S sites) [32]. The α–SrZrS_3_ structure consists of one–dimensional double chains of edge–sharing ZrS_6_ octahedra along the b axis, with Sr atoms in nine–fold coordination with S atoms, forming tri–capped trigonal prisms (Figure 1a) that are interconnected via common triangular faces [32]. Based on powder neutron diffraction analysis, the β–SrZrS_3_ is said to have a distorted perovskite structure [33]. Unlike the α–phase, the structure of β–SrZrS_3_ is constituted by three–dimensionally connected corner–sharing Zr octahedra, and Sr atoms which are eightfold coordinated in a bi-capped trigonal prism (Figure 1b). Summarized in Table 1 are the calculated lattice parameters obtained from a full structure optimization at the using PBE and HSE06 functionals, which show good agreement with known experimental data.^32^ In the α–phase, each of the ZrS_6_ octahedra shares two opposite edges with one another, forming a linear chain which in turn, interconnect in pairs through additional edge–sharing. The Zr–S bonds form the shortest, having a length of 2.45 Å, whereas the longest bond having a length of 2.65 Å, which involves an S atom bonding to three Zr atoms, occurs opposite to the shortest bond. The Sr–S bonds in the α–SrZrS_3_ phase ranges from 3.06–3.09 Å. Conversely, the ZrS_6_ octahedra of the β–phase have all their corners interconnected forming a three–dimensional network, with Zr–S bonds ranging from 2.54 to 2.57 Å, in close agreement with experimental values (Table 1).

Based on the optimized structures, we have simulated X–ray diffraction (XRD) spectrum of α–SrZrS_3_ and β–SrZrS_3_ using the VESTA Crystallographic Software as shown in Figure 2a. We can see that clear differences between the assigned peaks of α–SrZrS_3_ and β–SrZrS_3_, which is consistent with the difference in their lattice parameters. All the peaks in the simulated DFT spectrum match very closely with the experimental XRD measurement from the work of Niu et al. [34], as shown (Figure 2b). We consider that the assigned reflection peaks in the DFT XRD spectrum may become useful in clarifying future experiments, for instance to distinguish between the α–SrZrS_3_ and β–SrZrS_3_ phases.

### 3.2. Mechanical Properties

The elastic properties of materials give the data necessary in understanding the bonding property between adjacent atomic planes, stiffness, bonding anisotropic and structural stability of the material [35,36]. Shown in Table 2 are the calculated single crystal elastic constants of the α– and β–SrZrS_3_ materials, all of which satisfy the Born’s mechanical stability criteria for orthorhombic structures [37], indicating that both materials are mechanically stable under ambient conditions. The predicted trend of C_11_ > C_22_ > C_33_ for the α–SrZrS_3_ phase, indicates greater stiffness in the [100] direction than in the [010] and [001] directions. For the β–SrZrS_3_ phase, C_22_ > C_33_ > C_11_ which implies that β–SrZrS_3_ is stiffest in the [010] direction and least stiff in the [100] direction. The high elastic stiffness of the α– and β–SrZrS_3_ could be attributed to strong Sr–S and Zr–S chemical bonding [35]. Summarized in Table 2 are the calculated Bulk (B), shear (G) and Young’s (E) moduli. A higher bulk modulus is predicted for the β–SrZrS_3_ phase (79.9 GPa), similar to the value of 77.35 predicted by Oumertem et al.^15^, compared to the α–SrZrS_3_ (61.7 GPa) phase. Similar bulk modulus values were obtained by fitting a third–order Birch–Murnaghan (3^rd^ BM) equation of state (EOS) [38,39] to the DFT–PBE obtained total electronic energy (E) vs unit cell volume (V) data (Figure 3) based on the equation,
(6)$$E\left(V\right)=E_{0}+\frac{9V_{0}B_{0}}{16}\left\{{\left[{\left(\frac{V_{0}}{V}\right)}^{\frac{2}{3}}-1\right]}^{3}B_{0}^{\prime }+{\left[{\left(\frac{V_{0}}{V}\right)}^{\frac{2}{3}}-1\right]}^{2}\left[6-4{\left(\frac{V_{0}}{V}\right)}^{\frac{2}{3}}\right]\right\}$$
where *E*_0_ and V_0_ are the equilibrium energy and volume, *B*_0_ is hydrostatic bulk modulus and B’ is the pressure derivative of the bulk modulus at T = 0 K and P = 0 GPa. The bulk modulus and its pressure derivative are calculated at 66.73 GPa and 2.33 for α–SrZrS_3_ and 83.75 GPa and 2.93 for β–SrZrS_3_, respectively. This result suggests that the β–SrZrS_3_ phase is more rigid and less prone to compressive deformation than the α–SrZrS_3_ phase. Conversely, the α–phase has higher shear and Young’s moduli than the β–phase implying more resistance to shear and tensile deformation. The Poisson’s ratios (*v*) are predicted at 0.244 and 0.436 for the α, and β–phases, respectively. The Poisson’s ratio allows us to test the ductility/brittleness of material. A material is characterized as ductile if *v* < 0.5, otherwise it is classified as brittle [40]. Based on this criterion, we conclude that both α– and β–SrZrS_3_ are ductile materials, which implies that any strain-induced defects at the interface, when these materials are deposited on substrates may relax over a relatively short distance. 

### 3.3. Electronic Properties

Shown in Figure 4a,b are the electronic band structures of the α–SrZrS_3_ and β–SrZrS_3_ phases, respectively, with the corresponding density of states projected on the Sr–*d*, Zr–*d* and S–*p* states in Figure 5a,b. Analysis of the band structures reveal that both α– and β–SrZrS_3_ are direct band gaps absorbers seeing the bottom of the conduction band and the top of the valence band are located at the same crystal momentum (Γ) points on the Brillouin zone. The band gap energies of α– and β–SrZrS_3_ are predicted at 1.38, and 1.95 eV, respectively. The predicted values are in close agreement with the estimated optical band gaps from diffuse–reflectance measurements: α–SrZrS_3_ (1.52 eV) and β–SrZrS_3_ (2.05 eV) [16]. It is evident from the projected density of states (Figure 5a,b) that the conduction band edge, in both phases, is dominated by Zr–*d* states. Whereas, the contribution from S–*p* orbital dominates the valence band edge.

The charge carrier effective masses (*m**) which is intricately linked to the diffusion coefficient (*D*) and mobility (*μ*) of charge carrier in a semiconductor via the relations $D=\frac{k_{B}T}{e}\mu$ and $\mu =e\frac{\tau }{m^{*}}$, were also calculated for the α– and β–SrZrS_3_ materials. Small photo-carrier effective masses result in increased *μ* and *D.* The conductivity of charge carrier effective mass dictates the electrical resistivity and optical response of solar devices; hence their calculation is important [41]. Whiles effective masses can be quite difficult to obtain experimentally, accurate DFT calculations [42,43] can complement experiments by computing these properties. The effective masses can be obtained by fitting the energy of the valence band maximum (holes) and conduction band minimum (electrons) to a quadratic polynomial in the reciprocal lattice vector *k* based on the equation$\;m_{e\left(h\right)}^{*}=\pm \hbar^{2}{\left(\frac{d^{2}E_{k}}{dk^{2}}\right)}^{-1}.\;$In Table 3 the calculated electron and hole effective masses for the α– and β–SrZrS_3_ materials are summarized in some selective directions of the Brillouin zone. The predicted smaller, and therefore, the light mass charge carriers correspond to a high mobility of the electrons and holes at the conduction, and valence bands, respectively, and consequently point to high conductivity. The high conductivity also demonstrates the efficient separation of photo-generated charge carriers, which give rise to high-efficiency fabricated solar cell devices. In general, we found that the holes have higher effective masses than electrons in both the α– and β–SrZrS_3_ materials, suggesting that the electrons do tunnel much readily than the holes. The large effective mass difference between the electrons and holes is an important factor in minimizing their recombination rate [44]. By computing the ratio of the hole to electron effective masses (*D* = m^∗^_h_/m^∗^_e_), the recombination rate of photo-generated charge carriers can be assessed [45]. Higher *D* values generally signify higher mobility and a lower recombination rate of the photo-generated charges [46,47]. As shown in Table 3, the highest D values for the α–SrZrS_3_ phase were calculated along the Y− Γ (26.67), T− Γ (5.25), Z−U (5.00), U−R (3.20), and Γ−X (2.75) directions. For the β–SrZrS_3_ phase, the highest D values were obtained along the Γ−X (13.50), Z−U (7.40), X−S (4.67), Γ−Z (3.40), Y− Γ (2.36), and U−R (2.33). The higher *D* values along various directions on the Brillion zone points to efficient separation and low recombination of photo-generated charge carriers in the α– and β–SrZrS_3_ materials, which is necessary for the fabrication of highly efficient solar device.

### 3.4. Optical Properties

The calculated real (dispersive, $\varepsilon_{1}$) and imaginary (absorptive, $\varepsilon_{2}$) parts of the dielectric function for α–SrZrS_3_ and β–SrZrS_3_ are shown in Figure 6 and Figure 7, respectively. The dielectric constant is predicted at 9.36 for α–SrZrS_3_ (Figure 6a) and 15.75 β–SrZrS_3_ (Figure 7a). The predicted high dielectric constants are desired properties for photovoltaics applications as a dielectric constant value of 10 or more is good enough to obtain exciton binding energy (*E*_b_) value lower than 25 meV at room temperature [48]. The absorbance of α–SrZrS_3_ (Figure 6b) starts at around 1.5 eV, close to its band gap, but with negligible absorption until after 2 eV. The absorption of β–SrZrS_3_ (Figure 7b) starts at around 2.0 eV, which corresponds to it fundamental band gap. Owing the orthorhombic crystal symmetry of α– and β–SrZrS_3_, the absorption coefficient, reflectivity and refractive index plotted along the three the crystallographic directions: x, y, and z directions ([100], [010], and [001]) are found to be anisotropic. The absorption coefficient of a material represents its light harvesting ability, which is very necessary as it has great effects on the power conversion efficiency of the resulting solar cells. The high absorption coefficient is desired in solar cell applications. The calculated absorption coefficient, reflectivity and refractive index are shown in Figure 6 and Figure 7. A high absorption coefficient in the order of 10^5^ cm^–1^ is predicted for both α– and β–SrZrS_3_ in the visible light region, which make them suitable for the photovoltaic application. The β–SrZrS_3_ phase shows stronger absorption around 2 eV than α–SrZrS_3_ phase. 

The reflectivity and refractive index are two important parameters necessary for solar applications. The reflectivity gives a measure of reflecting light or radiation. When it comes to solar cells, the less the material surfaces reflect a sun’s rays, the more energy can be generated. The refractive index of a material, on the other hand, shows its transparency. The optical reflectivity of α–SrZrS_3_ (Figure 6c) shows a high reflectivity in the y–direction, starts at about 30.2% and reaches a maximum value of about 68.2% at an energy of about 3.8 eV. The static reflectivity in the x, y, and z directions are predicted at about 23%, 30%, and 26%, respectively. This indicate that the least reflection occurs in the z direction. The calculated static reflectivity of β–SrZrS_3_ (Figure 7c) is predicted to be about 30%, 34% and 38% in the y, z, and x directions, respectively. The reflectivity reaches a maximum (82.2%) at an energy of about 3.5 eV. The high reflectivity in the visible region could cause a significant loss in solar cell efficiency of devices fabricated. Optical losses from reflected light from the front surface can be reduces through surface texturing and anti–reflection coatings. The refractive index is predicted in the range of 2.8–3.5 for α–SrZrS_3_ (Figure 6d) and 3.5–4.0 for β–SrZrS_3_ (Figure 7d). The calculated refractive indexes for α– and β–SrZrS_3_ are similar to the refractive index of Si (η = 3.4 at 550 nm) [49]. 

## 4. Summary and Conclusions 

We performed comprehensive first–principles GGA and hybrid DFT investigations the structures and properties α–SrZrS_3_ and β–SrZrS_3_ transition metal chalcogenide perovskites. Both α– and β–SrZrS_3_ materials were demonstrated to be mechanically stable at ambient conditions, based on their calculated single–crystal elastic constants. The predicted electronic structures show that both α– and β–SrZrS_3_ are direct band gaps absorbers with band gap of calculated at 1.38, and 1.95 eV, respectively, in close agreement with estimates from diffuse–reflectance measurements. Based on the dielectric functions obtained, we show that α– and β–SrZrS_3_ have strong light absorption in the visible region, as reflected in their high optical absorbance (in the order 10^5^ cm^−1^). The β–SrZrS_3_ shows a stronger absorption around 2 eV than α–SrZrS_3_. The first theoretical prediction of effective masses of photo-generated charge carriers, in α– and β–SrZrS_3_ materials, suggest s high-mobility (high-conductivity) and low recombination rate of photo-carriers in these materials, making them attractive for solar cell and other optoelectronic applications.

## Figures and Tables

**Figure 1 materials-13-00978-f001:**
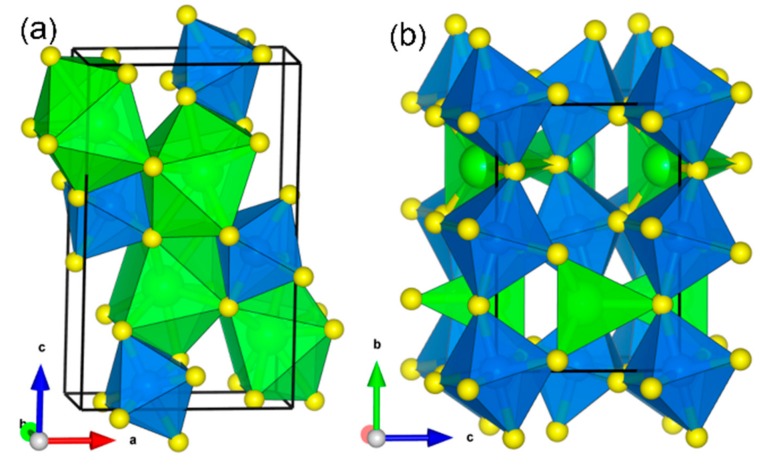
Crystal structure of (**a**) α–SrZrS_3_ and (**b**) β–SrZrS_3._ Color code: Sr = Green, Zr = Blue, and S = yellow.

**Figure 2 materials-13-00978-f002:**
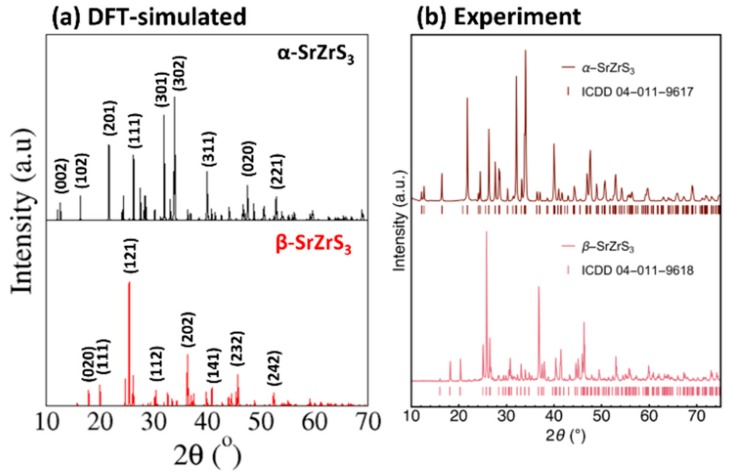
(**a**) DFT–simulated X–ray diffraction pattern of α–SrZrS_3_ and β–SrZrS_3_ compared with (**b**) experimental XRD measurement [34].

**Figure 3 materials-13-00978-f003:**
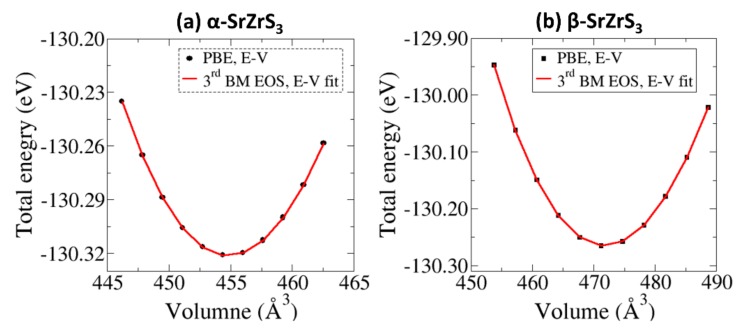
The third–order Birch–Murnaghan equation of state (EOS) fitting to the calculated E−V data of (**a**) α–SrZrS_3_ and (**b**) β–SrZrS_3_.

**Figure 4 materials-13-00978-f004:**
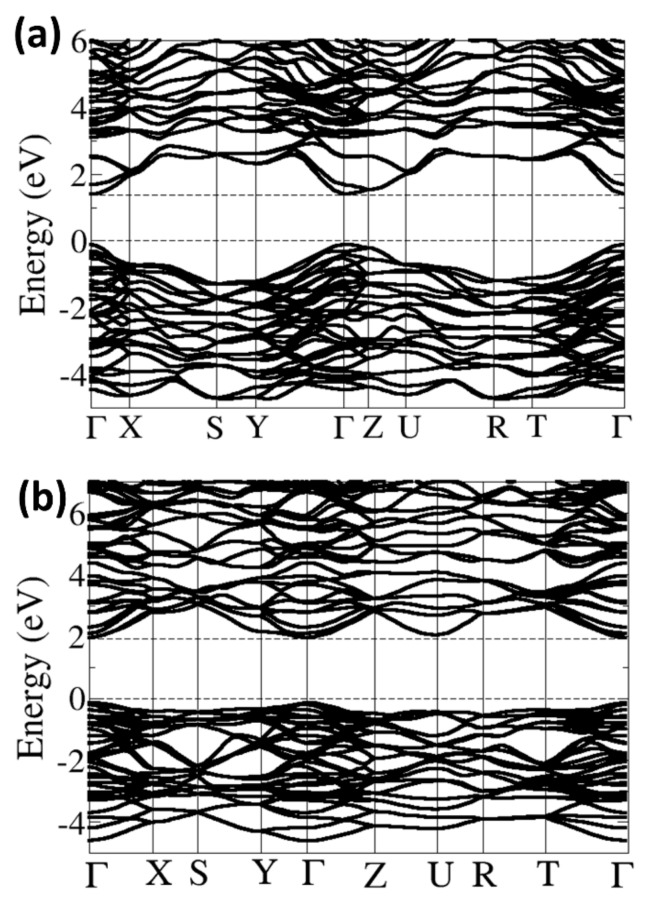
Band structure along the high–symmetry directions of the Brillouin zone of (**a**) α–SrZrS_3;_ and (**b**) β–SrZrS_3_.

**Figure 5 materials-13-00978-f005:**
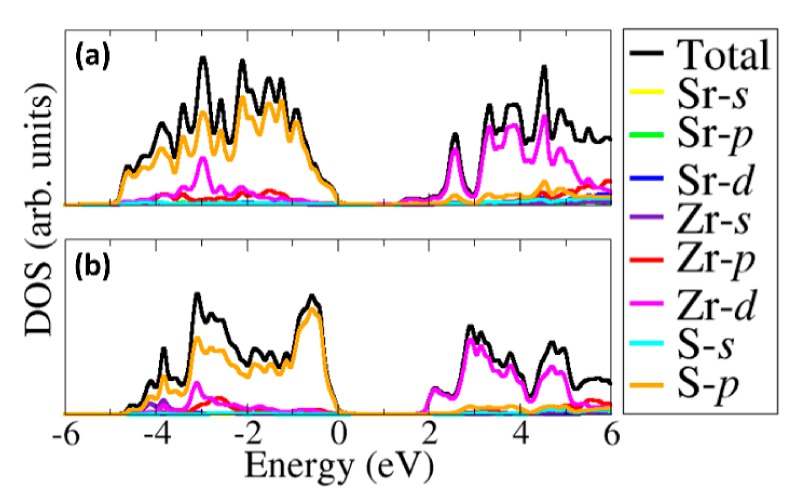
The partial density of states for (**a**) α–SrZrS_3_; and (**b**) β–SrZrS_3_.

**Figure 6 materials-13-00978-f006:**
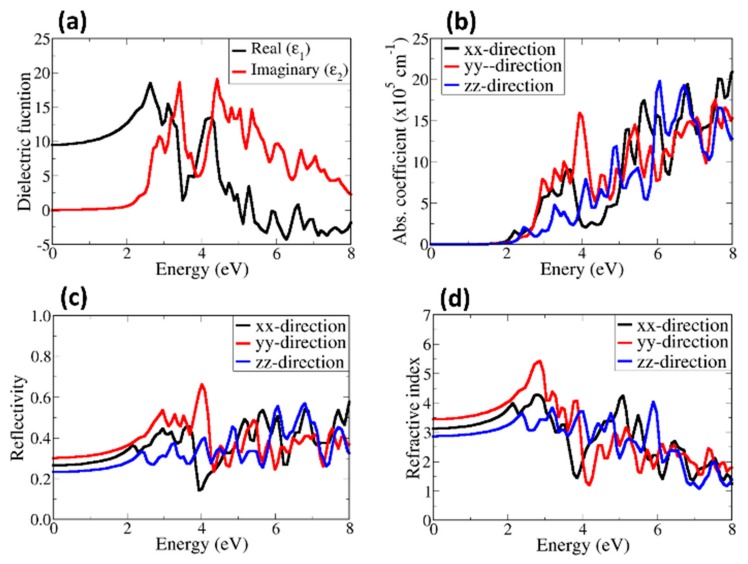
Calculated (**a**) dielectric function, (**b**) absorbance, (**c**) reflectivity and (**d**) refractive index of α–SrZrS_3_.

**Figure 7 materials-13-00978-f007:**
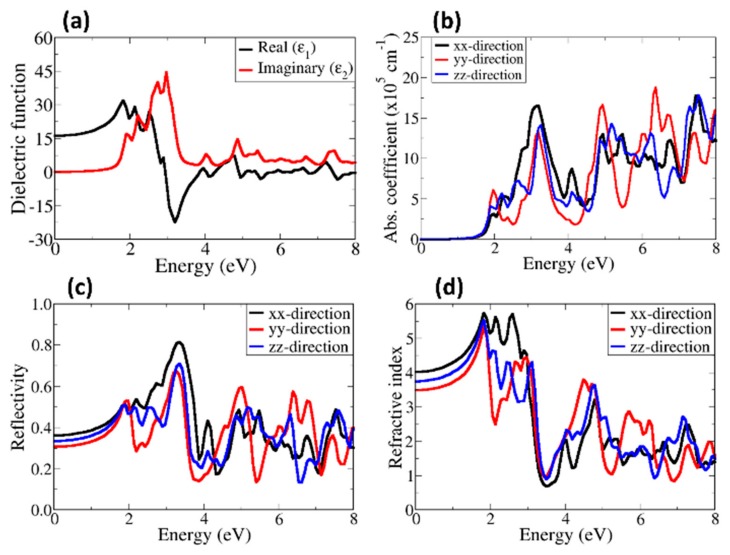
Calculated (**a**) dielectric function, (**b**) absorbance, (**c**) reflectivity and (**d**) refractive index of β–SrZrS_3_.

**Table 1 materials-13-00978-t001:** Lattice constant (Å) and bond lengths (Å) of for α–SrZrS_3_ and β–SrZrS_3_.

	α–SrZrS_3_			β–SrZrS_3_		
Parameter	PBE–D3	HSE06+D3	Experiment [32]	PBE+D3	HSE06+D3	Experiment [32]
a (Å)	8.551	8.540	8.525	7.133	7.125	7.109
b (Å)	3.814	3.810	3.826	9.783	9.772	9.766
c (Å)	13.930	13.908	13.925	6.752	6.748	6.735
d(Sr–S)	3.06–3.09	3.05–3.09	3.06–3.08	2.99–3.16	2.98–3.21	2.96–3.37
d(Zr–S)	2.45–2.65	2.44–2.63	2.44–2.64	2.54–2.57	2.54–2.56	2.53–2.56

**Table 2 materials-13-00978-t002:** Elastic constants (C_ij_), bulk modulus (B), shear modulus (G), Young’s modulus (E), and Poisson’s ratio (*v*) of α–SrZrS_3_ and β–SrZrS_3_.

Parameter	α–SrZrS_3_	β–SrZrS_3_
C_11_	137.3	50.3
C_22_	107.4	242.3
C_33_	74.3	118.5
C_44_	32.0	50.9
C_55_	35.4	29.3
C_66_	56.3	8.9
C_12_	44.4	53.1
C_13_	63.6	34.9
C_23_	10.3	66.0
B (GPa)	61.73	79.9
G (GPa)	38.12	34.96
E (GPa)	95.36	91.53
υ	0.244	0.436

**Table 3 materials-13-00978-t003:** Calculated hole (m^∗^_h_) and electron (m^∗^_h_) effective masses of α–SrZrS_3_ and β–SrZrS_3_ along high symmetry directions.

Material	Direction	m^∗^_h_ (m_e_)	m^∗^_e_ (m_e_)	D= m^∗^_h_/m^∗^_e_
α–SrZrS_3_	Γ−X	0.011	0.004	2.75
	X−S	0.010	0.007	1.43
	S−Y	0.015	0.013	1.15
	Y− Γ	0.080	0.003	26.67
	Γ−Z	0.007	0.005	1.40
	Z−U	0.020	0.004	5.00
	U−R	0.016	0.005	3.20
	R−T	0.023	0.049	0.47
	T− Γ	0.021	0.004	5.25
β–SrZrS_3_	Γ−X	0.054	0.004	13.50
	X−S	0.014	0.003	4.67
	S−Y	0.017	0.067	0.25
	Y− Γ	0.026	0.011	2.36
	Γ−Z	0.017	0.005	3.40
	Z−U	0.037	0.005	7.40
	U−R	0.007	0.003	2.33
	R−T	0.012	0.015	0.80
	T− Γ	0.009	0.008	1.13

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
