# Peer review of "First–Principles Investigation of the Structural, Elastic, Electronic, and Optical Properties of α– and β–SrZrS3: Implications for Photovoltaic Applications"

_materials, 2020, doi:10.3390/ma13040978_

Round 1

Reviewer 1 Report

This manuscript presents interesting results on a new kind of 2D monolayers of carbon selenide. The authors employed a standard DFT methodology to investigate the the structural, electronic, and optical properties of pristine chalcogenide perovskites. This manuscript deserves to be published in Materials provided that major and minor revisions are made by the authors.

Major comments:

1. The manuscript is well written and easy to follow its information.

2. Are these compounds p-type or n-type semiconductors? Extrinsic or intrinsic ones?

Minor comments:

1. Replace Table 2 by Table in line 154

2. It will be very interesting to the audience to include in Tables 1 and 2 the lattice constants, bond lengths, and elastic constants obtained with PBE to see what advantages does HSE06 bring when optimising the geometry.

3. Could the authors provide also the fundamental band gap as the difference between the ionisation energy and electron affinity? It will be interesting to see the differences with the other gaps reported from experiments and DFT/HSE06 calculations.

4. Please fix the error reference in line 253.

Reviewer 2 Report

The manuscript reports an accurate theoretical study of alpha- and beta- phases of SrZrS_3, showing that these materials can be useful for photovoltaic applications,
such as solar cells. The authors performed Density Functional Theory calculations, with an accurate PBE-D3 exchange-correlation functional for geometry optimization, and short-range screened hybrid HSE06 functional for band structure and
optical properties.

Overall, the manuscript is interesting and may be important for optical applications. But before publication, I have some minor points to be addressed:

- The authors should mention how accurate is the independent-particle formalism used for optical calculations. Can this method show correctly the excitonic effects? More expensive methods, such as BSE and time-dependent DFT with proper exchange-correlation kernels (e.g. see Phys. Rev. B 98, 085123 (2018)) should be briefly mentioned and cited.

- the bulk modulus has been calculated from elastic constants using Eq. (4). But usually it is computed from equation of states (such as Murnaghan, Birch, etc). A
comparison between these methods can be provided in the manuscript.

some typos:

lines 241 and 253: ...are shown in Error! Reference source not found.

line 244: ...values of 10 or more -> ...value of 10 or more

lines 241-242: The dielectric constant is predicted at 9.36 ...
